# Malaria-associated atypical memory B cells exhibit markedly reduced B cell receptor signaling and effector function

Silvia Portugal[1], Christopher M Tipton[2,3], Haewon Sohn[1], Younoussou Kone[4], Jing Wang[1], Shanping Li[1], Jeff Skinner[1], Kimmo Virtaneva[5], Daniel E Sturdevant[5], Stephen F Porcella[5], Ogobara K Doumbo[4], Safiatou Doumbo[4], Kassoum Kayentao[4], Aissata Ongoiba[4], Boubacar Traore[4], Inaki Sanz[2], Susan K Pierce[1], Peter D Crompton[1]*

[1]Laboratory of Immunogenetics, National Institute of Allergy and Infectious Diseases, National Institutes of Health, Rockville, United States; [2]Departments of Medicine, Division of Rheumatology, Lowance Center for Human Immunology, Emory University, Atlanta, United States; [3]Department of Pediatrics, Division of Rheumatology, Lowance Center for Human Immunology, Emory University, Atlanta, United States; [4]Malaria Research and Training Centre, Department of Epidemiology of Parasitic Diseases, International Center of Excellence in Research, University of Sciences, Technique and Technology of Bamako, Bamako, Mali; [5]Rocky Mountain Laboratory Research Technologies Section, Genomics Unit, National Institute of Allergy and Infectious Diseases, National Institutes of Health, Hamilton, United States

*For correspondence:
pcrompton@niaid.nih.gov

Competing interests: The authors declare that no competing interests exist.

**Abstract** Protective antibodies in *Plasmodium falciparum* malaria are only acquired after years of repeated infections. Chronic malaria exposure is associated with a large increase in atypical memory B cells (MBCs) that resemble B cells expanded in a variety of persistent viral infections. Understanding the function of atypical MBCs and their relationship to classical MBCs will be critical to developing effective vaccines for malaria and other chronic infections. We show that VH gene repertoires and somatic hypermutation rates of atypical and classical MBCs are indistinguishable indicating a common developmental history. Atypical MBCs express an array of inhibitory receptors and B cell receptor (BCR) signaling is stunted in atypical MBCs resulting in impaired B cell responses including proliferation, cytokine production and antibody secretion. Thus, in response to chronic malaria exposure, atypical MBCs appear to differentiate from classical MBCs becoming refractory to BCR-mediated activation and potentially interfering with the acquisition of malaria immunity.

## Introduction

*Plasmodium falciparum* is a mosquito-born parasite that causes approximately 200 million cases of malaria and 600,000 deaths each year, mostly among African children (*WHO, 2014*). The development of a highly effective vaccine is widely viewed as a critical step toward defeating malaria, yet the vaccine candidate that is most advanced in clinical trials confers only partial, short-lived protection in African children (*RTS, S Clinical Trials Partnership, 2014*).

Abs play a key role in naturally acquired immunity to malaria as demonstrated by the passive transfer of Abs from malaria-resistant adults to children with clinical malaria, resulting in a reduction in the levels of parasitemia and fever in these children (*Cohen et al., 1961*). Individuals living in malaria endemic areas acquire protective Abs but the process is remarkably slow requiring many years of repeated *P. falciparum* infections (*Portugal et al., 2013*). The inefficient acquisition of humoral

**eLife digest** The human immune system works to protect individuals from harmful microbes, such as the parasites that cause malaria. One line of defense is to produce a large array of proteins called antibodies that specifically bind to microbes to mark them for destruction by the immune system. The immune system also produces long-lived memory B cells that are able to mount a quicker and more effective antibody response if the microbe enters the body again. This means that most people only become ill with a particular disease the first time they encounter the microbe that causes it.

However, malaria is unusual in that it can take many years of exposure to the parasite that causes it before an individual produces enough antibodies and memory B cells to be protected from the disease. There is also no vaccine that provides effective and long-lasting protection against malaria. Vaccinations rely on stimulating the body's natural defenses, and so understanding more about antibodies and memory B cells in relation to malaria may aid future efforts to develop a vaccine.

Researchers have discovered that many of the memory B cells that accumulate in people who have been exposed to the malaria parasite over long-periods of time are different from the normal memory B cells. But it was not clear what role these 'atypical' cells play in immunity to malaria. To address this question, Portugal et al. studied the genetics and activity of B cells collected from children and adults living in Mali who—by living in a region where malaria is common—had been repeatedly exposed to the parasite.

The experiments indicate that atypical and normal memory B cells both develop from the same precursor cells. However, the genes that are active in each cell type are different, resulting in the atypical cells being less able to respond to the parasite than the normal memory B cells. Portugal et al.'s findings suggest that the atypical cells develop from normal memory B cells during long-term exposure to malaria, which may delay the development of immunity to this disease.

Future challenges include understanding what drives the formation of the atypical memory B cells in malaria, and finding out why they are less active than the normal cells. This could aid the development of vaccines and/or therapies that restore their activity in patients.

immunity that protects from malaria has been attributed, in part, to the extensive genetic diversity of *P. falciparum* parasites (*Takala and Plowe, 2009*) and the extraordinary clonal variation in the proteins the parasite expresses on the surface of the erythrocytes that it infects (*Scherf et al., 2008*). However, accumulating evidence suggests that *P. falciparum* may also evade humoral immunity through dysregulation of B cell responses (*Portugal et al., 2013*; *Scholzen and Sauerwein, 2013*; *Hviid et al., 2015*). Indeed, several studies, particularly in children, show that *P. falciparum*-specific Ab responses are short-lived (*Muller et al., 1989*; *Fruh et al., 1991*; *Ramasamy et al., 1994*; *Taylor et al., 1996*; *Cavanagh et al., 1998*; *Fonjungo et al., 1999*; *John et al., 2002*; *Akpogheneta et al., 2008*; *Weiss et al., 2010*) and boost inconsistently upon antigen re-exposure (*Kinyanjui et al., 2007*), even to conserved antigens (*Fonjungo et al., 1999*). Moreover, in areas of intense malaria transmission, *P. falciparum*-specific MBCs are acquired inefficiently (*Weiss et al., 2010*) and their prevalence is relatively low (~30–50%) even among adults with documented malaria exposure (*Dorfman et al., 2005*; *Weiss et al., 2010*; *Wipasa et al., 2010*; *Nogaro et al., 2011*; *Ndungu et al., 2012*).

Altered B cell function is well described in other infections. For example, HIV and hepatitis C virus infection are associated with an increase in CD10$^-$CD19$^+$CD20$^+$CD21$^-$CD27$^-$ B cells that express the inhibitory molecule Fc receptor-like-4 (FcRL4) (*Charles et al., 2008*; *Moir et al., 2008*), cells that are largely absent in the blood of healthy individuals. In HIV-infected individuals, these cells are hyporesponsive or 'exhausted', and because HIV-specific B cells are concentrated in this sub-population (*Kardava et al., 2014*), they likely contribute to the humoral deficiencies associated with HIV infection (*Moir et al., 2008*).

An increase in a phenotypically similar subset of B cells has been described in malaria-exposed children and adults across genetically and geographically diverse populations (*Weiss et al., 2009*; *Nogaro et al., 2011*; *Weiss et al., 2011*; *Portugal et al., 2012*; *Illingworth et al., 2013*; *Muellenbeck et al., 2013*). In the context of malaria, this B cell subset is referred to as 'atypical' (*Weiss et al., 2009*) rather than 'exhausted' because the function of these cells and whether they are beneficial or

detrimental in malaria remains unclear. That *P. falciparum* infection per se drives the expansion of atypical MBCs has been suggested by a positive correlation between atypical MBC expansion and *P. falciparum* transmission intensity (*Weiss et al., 2011*), the differential expansion of atypical MBCs in age-matched children living under similar conditions in rural Kenya, with the exception of *P. falciparum* exposure (*Illingworth et al., 2013*) and the appearance of atypical MBCs in the peripheral blood of healthy adults following experimental *P. falciparum* infection (*Scholzen et al., 2014*).

B cell memory is complex and encompasses distinct classes of MBCs, and at present the origins and functions of these MBC subsets are incompletely understood (*Tarlinton and Good-Jacobson, 2013*). In particular, in malaria the function of atypical MBCs and their relationship to classical MBCs remains to be established. Concerning function, *Muellenbeck et al. (2013)* recently showed that $V_H$ and $V_L$ genes cloned from atypical MBCs from malaria exposed adults encoded broadly neutralizing *P. falciparum*-specific Abs. This observation led these authors to speculate that atypical MBCs contribute to malaria immunity by secreting Abs against blood-stage *P. falciparum* parasites, although Ab secretion by atypical MBCs was not directly demonstrated. Concerning the relationship between atypical and classical MBCs, two recent analyses of the $V_H$ and $V_L$ sequences of atypical and classical MBC led to different conclusions. A study in Gabon reported that classical and atypical MBCs were different in their expressed IgG V gene repertoires suggesting that they developed from different precursors (*Muellenbeck et al., 2013*). In contrast, results from a more recent study in Mali indicated that the expressed IgG V gene repertoires of atypical and classical MBCs were remarkably similar suggesting a close relationship between the two populations (*Zinocker et al., 2015*). However, a relatively small number of V genes were analyzed in these two studies leaving the question of the relatedness of atypical and classical MBCs an open one.

Here, we sought to fill these important knowledge gaps by analyzing naïve B cells, classical MBCs and atypical MBCs isolated from Malian children and adults with lifelong *P. falciparum* exposure. Using next-generation sequence analysis of $V_H$ genes, we provide evidence that atypical MBCs share a common precursor with classical MBCs, on the basis of similar somatic hypermutation (SHM) rates and $V_H$ gene usage. By genome-wide expression profiling, we demonstrate that atypical MBCs upregulate multiple inhibitory receptors, particularly the Fc receptor-like-5 (FcRL5). Moreover, atypical MBCs exhibit reduced B cell receptor (BCR)-mediated signaling resulting in markedly impaired BCR-triggered $Ca^{2+}$ mobilization, proliferation and cytokine production. We show that atypical MBCs do not actively secrete Abs or differentiate into Ab-secreting cells (ASCs) in vitro. Thus, while atypical and classical MBCs appear closely related, atypical MBCs exhibit markedly reduced signaling and effector function, which may contribute to the inefficient acquisition of humoral immunity to malaria, findings that point toward the possibility of modulating B cell responses to enhance malaria vaccine efficacy in endemic areas.

## Results

### Atypical MBCs are greatly expanded in malaria-exposed individuals

In this study we analyzed naïve B cells (CD10⁻CD19⁺CD20⁺CD21⁺CD27⁻), classical MBCs (CD10⁻CD19⁺CD20⁺CD21⁺CD27⁺) and atypical MBCs (CD10⁻CD19⁺CD20⁺CD21⁻CD27) (*Figure 1A, B*) from the peripheral blood of 107 Malian adults with lifelong exposure to intense *P. falciparum* transmission, and 6 Malian children during acute febrile malaria. The average age of the adult subjects was 41.2 years (range 18–61) and 48% were female. The average age of the pediatric subjects was 8.1 years and 31% were female. Consistent with prior studies conducted in Mali and other malaria-endemic areas (*Weiss et al., 2009*; *Nogaro et al., 2011*; *Weiss et al., 2011*; *Portugal et al., 2012*; *Illingworth et al., 2013*; *Muellenbeck et al., 2013*), atypical MBCs in the adult subjects comprised 6–51% of total mature circulating B cells (mean 17.1% ± 8.2) compared to 0.6–8% (mean 2.9% ± 1.9) in malaria-naïve U.S. adults (p < 0.0001; *Figure 1C*).

To gain insight into the developmental history, phenotype and function of atypical MBCs and their relationship to naïve B cells and classical MBCs, we analyzed B cell subpopulation isolated from Malian adults and children using a variety of genomics-based, biochemical and functional assays.

### Atypical and classical MBCs share similar isotype distributions, replication histories and Ig repertoire features

At present, the factors that drive the differentiation of atypical MBCs are poorly understood. To gain insight into the developmental history of atypical MBCs and their relationship to naïve B cells and

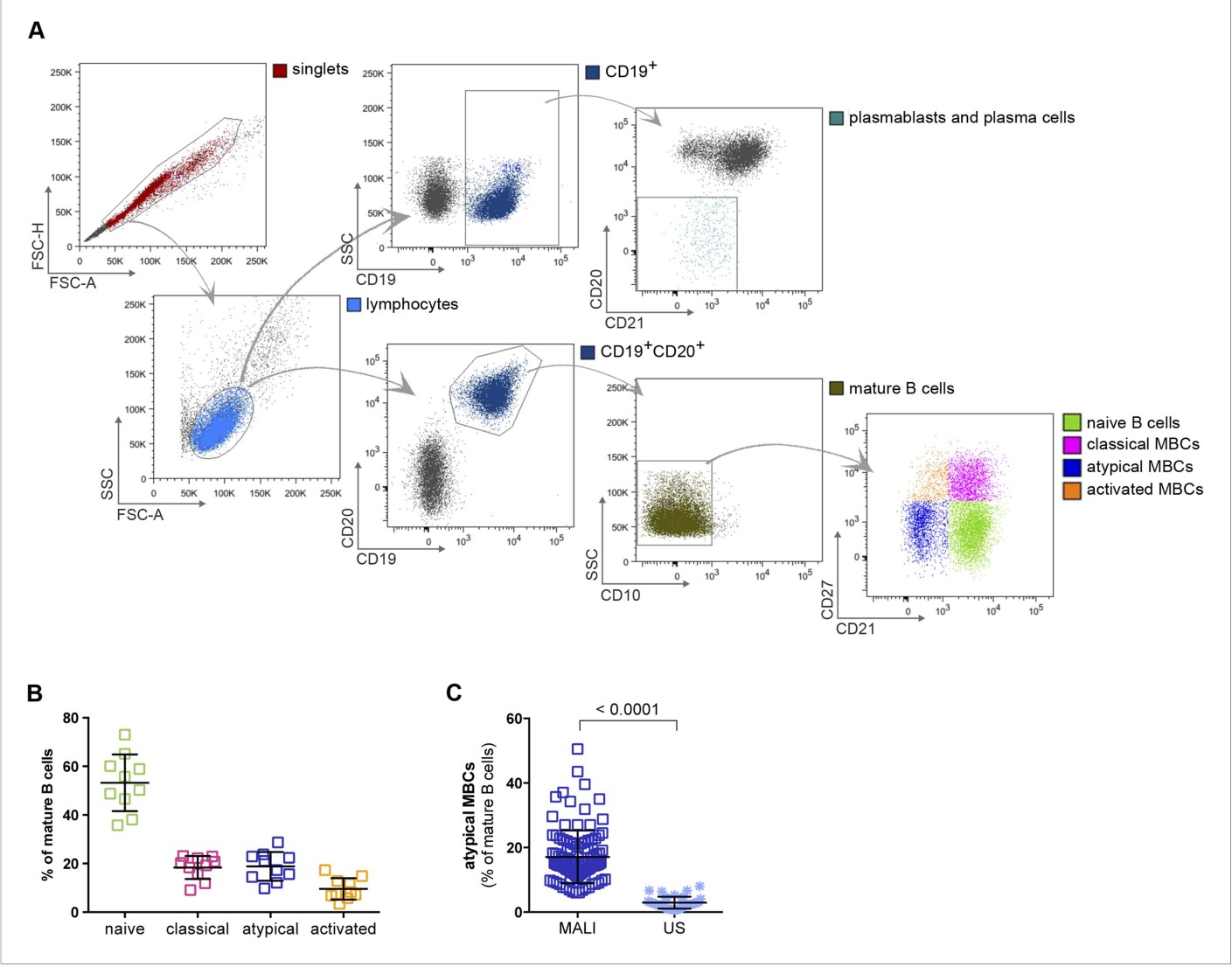

**Figure 1**. Atypical MBCs are markedly increased in individuals exposed to intense malaria transmission. (**A**) Flow cytometry gating strategy to identify B cell subpopulations in a representative Malian adult. (**B**) Distribution of naïve B cells (CD10⁻CD19⁺CD20⁺CD21⁺CD27⁻), classical MBCs (CD10⁻CD19⁺ CD20⁺CD21⁺CD27⁺), atypical MBCs (CD10⁻CD19⁺CD20⁺CD21⁻CD27⁻) and activated MBCs (CD10⁻CD19⁺CD20⁺CD21⁻CD27+) as a percentage of total mature B cells in representative Malian adults (n = 10 subjects). (**C**) Atypical MBCs as a percentage of total mature B cells in Malian adults (n = 107) and U. S. adults (n = 37). Horizontal bars and whiskers represent means and SD, respectively. p value determined by unpaired ttest.

classical MBCs, we first compared their Ig isotype distributions by flow cytometry. As expected, naïve B cells were largely IgM⁺ while classical and atypical MBCs shared an Ig isotype profile that was predominantly IgG⁺ (*Figure 2A*). To determine the in vivo replication histories of the three B cell subpopulations, B cells were purified from peripheral blood mononuclear cells by removing all non-B cells and plasma cells/plasmablast yielding B cells of >98% purity. B cells were then FACS-sorted to obtain naïve B cells, classical and atypical MBCs (*Figure 2B* and *Figure 2—figure supplement 1*) and the replication histories were determined using a qRT-PCR-based assay that measures κ-deletion recombination excision circles (KRECs) (*van Zelm et al., 2007*). KRECs are produced in B cells during Ig gene rearrangements in the bone marrow. When B cells exit the bone marrow, each subsequent cell division dilutes the number of KRECs by a factor of two, allowing the number of in vivo cell divisions to be estimated. We found the average number of cell divisions to be similar between atypical and classical MBCs (atypical mean 5.4 ± 0.9; classical mean 6.5 ± 0.9; p = 0.24), whereas the number of cell divisions was predictably low in naïve B cells (mean 0.3 ± 0.7) (*Figure 2C*).

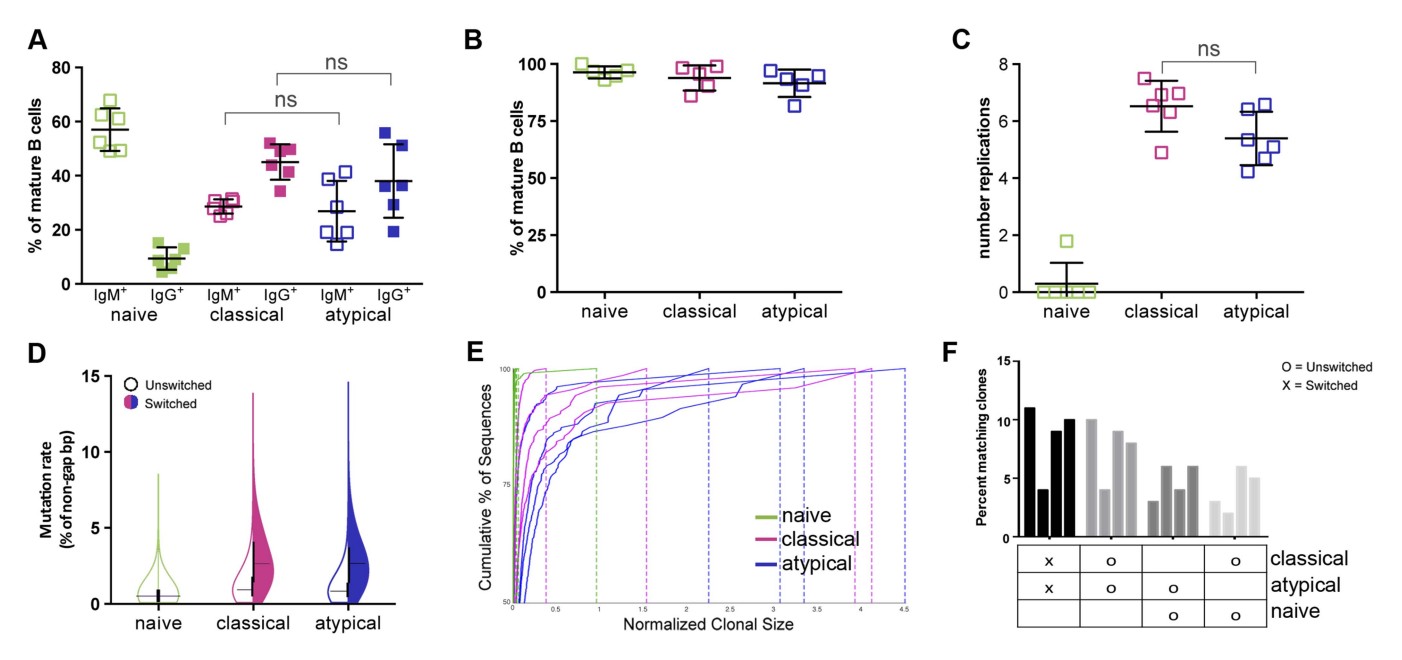

**Figure 2**. Atypical and classical MBCs are similar in isotype distribution, replication history and Ig repertoire characteristics. (**A**) Flow cytometry data showing the distribution of IgM+ and IgG+ cells as a percentage of each B cell subpopulation in Malian adults (n = 7 subjects). (**B**) Purity of B cell subpopulations as a percentage of mature B cells following FACS sorting. (**C**) Average number of in vivo replications determined by KREC assay for each B cell subpopulation in Malian adults (n = 6 subjects). (**D**) Somatic hypermutation rates of unswitched and switched B cell subpopulations based on IgH sequence analysis. Shown is average of 4 Malian adults. (**E**) Clonality of B cell subpopulations determined by IgH sequence analysis and displayed by size-ranking clonal lineages from top (largest) to bottom (smallest) along the y-axis representing 50% of all sequences from the largest clones. X-axis is the normalized lineage size of each clone (percentage of the total number of sequences). Shown is data from 4 Malian adults. (**F**) Percentages of IgH sequences in both unswitched and switched B cell subpopulations that are shared by other subpopulations. Each bar represents a different subject (n = 4 subjects). Horizontal bars and whiskers represent means and SD, respectively. Horizontal bars on (**D**) represent means. p values determined by ANOVA with Tukey's multiple comparisons test (**A** and **C**).

The following figure supplements are available for figure 2:

**Figure supplement 1**. Flow cytometry gating strategy to determine purity of naïve B cells (CD10−CD19+CD20+CD21+CD27−) (**A**), classical MBCs (CD10−CD19+CD20+CD21+CD27+) (**B**), and atypical MBCs (10−CD19+CD20+CD21−CD27−) (**C**) after flow cytometry sorting.

**Figure supplement 2**. Bioinformatics analysis of NGS data.

To further understand how antigen-driven selective pressures influence the acquisition of atypical MBCs, we magnetically isolated naïve B cells, classical and atypical MBCs from purified B cells obtained from uninfected adults (blood smear negative), sequenced their rearranged Ig heavy chains and compared their SHM rates and clonal diversity profiles. As expected, naïve B cell $V_H$ had low SHM rates, whereas classical and atypical MBC $V_H$ had similar SHM rates of approximately 3% for isotype switched clones (*Figure 2D* and *Figure 2—figure supplement 2*). Also as expected, naïve B cells were highly polyclonal, whereas classical and atypical MBCs had similar clonal expansion profiles (*Figure 2E*) and shared ~10% of their clones (*Figure 2F*).

Thus, atypical and classical MBCs share similar isotype distributions, replication histories and Ig repertoire characteristics, which taken together, suggest that atypical and classical MBCs are derived from a common precursor and differentiate in response to similar antigen-dependent selective pressures.

## Atypical MBCs upregulate inhibitory receptors and differentially express BCR and co-stimulatory signaling molecules

To investigate the phenotype and function of malaria-associated atypical MBCs we performed genome-wide expression profiling of purified B cells magnetically sorted to obtain naïve B cells

(CD19+CD21+CD27−), classical MBCs (CD19+CD21+CD27+) and atypical MBCs (CD19+CD21−CD27−) from a subset of 20 uninfected Malian adults (blood smear negative) with lifelong exposure to intense malaria transmission. The average age of these subjects was 33.5 years (range 18–47) and 60% were female. Following magnetic sorting the purity of naïve B cells, classical and atypical MBCs was verified by flow cytometry using CD19, CD21 and CD27 specific monoclonal Abs. The purity achieved was 93.2% (95% CI: 91.0–95.4) 73.0% (95% CI: 68.7–77.4) and 81.9% (95% CI: 75.4–88.5) respectively (*Figure 3A* and *Figure 3—figure supplement 1*). Principal components analysis of the microarray data showed segregation of transcription profiles based on B cell subpopulation, but not age, gender or batch effects (*Figure 3B*). Within-subject differences in gene-expression were computed and resulted in 1115 differentially expressed genes (DEGs) between atypical MBCs and classical MBCs; 1082 DEGs between atypical MBCs and naïve B cells; and 161 DEGs between classical MBCs and naïve B cells (*Supplementary file 1*). Ingenuity Pathway Analysis (IPA) identified 'PI3K Signalling in B Lymphocytes' (p = 1.11e-5), 'Phospholipase C Signalling' (p = 1.08e-4), 'BCR Signalling' (p = 4.98e-4) and 'ICOS-ICOSL Signalling' (p = 1.41e-3) among the top canonical pathways that distinguished classical from atypical MBCs (*Figure 3C*).

Consistent with the IPA analysis, atypical MBCs showed differential expression of several molecules involved in BCR and co-stimulatory signaling (*Figure 3D*). In addition, atypical MBCs upregulate several inhibitory receptors that are known to modulate B cell function including FcRL3 and FcRL5 (*Figure 3D*), members of the FCRL family that are preferentially expressed by B cells and possess tyrosine-based immunomodulatory function (*Li et al., 2014*). Other inhibitory receptors significantly upregulated by atypical MBCs included CD72, CD200R1, LILRB1, LILRB2 and FCGR2B (*Figure 3D*). The differential expression of selected genes by microarray (CD35 [CR1], CCR7, IL4R, FGR, LILRB2, CD11c [ITGAX], CD200R1, ALOX5, CXCR5 AND FCRL5) was validated by quantitative real time (qRT)-PCR (r = 0.893; p < 2.30E-16) (*Figure 2E* and *Supplementary file 2A–C*).

Of note, FcRL4, an inhibitory receptor that is upregulated on tissue-like MBCs and HIV-associated exhausted MBCs (*Ehrhardt et al., 2005*; *Moir et al., 2008*), was not differentially expressed by atypical MBCs. Given the significance of FcRL4 as a marker for tissue-like and HIV-associated exhausted MBCs, we confirmed at the mRNA and protein levels by qPCR and flow cytometry, respectively, that malaria-associated atypical MBCs do not express FcRL4 (*Figure 3F,G*), whereas FcRL5, an inhibitory protein prominently upregulated by atypical MBCs in the microarray experiment (*Figure 3D*), was confirmed to be upregulated by atypical MBCs at the mRNA and protein levels by qPCR and flow cytometry, respectively (*Figure 3H,I*). Interestingly, the surface expression of FcRL5 protein by atypical MBCs was bimodal, with approximately 40% of atypical MBCs expressing FcRL5 on average (*Figure 3I*).

Together these data suggest that atypical and classical MBCs are programmed to respond differently to co-stimulatory signals and BCR engagement by antigen.

## Atypical MBCs exhibit diminished BCR signaling, Ca²⁺ mobilization, proliferation and cytokine production

Engagement of BCRs by antigen triggers the phosphorylation of immunoreceptor tyrosine-based activation motifs (ITAMs) in the cytoplasmic domains of the BCR's subunits Igα and Igβ by the src-family kinase Lyn (*Kurosaki and Hikida, 2009*). The phosphorylated ITAMs then recruit the cytosolic spleen tyrosine kinase (Syk) that is activated by phosphorylation by Lyn. Syk in turn phosphorylates and activates the B cell linker protein (BLNK), phosphoinositide 3-kinase (PI3K) and phospholipase Cγ2 (PLCγ2). When activated, these and other components of the proximal B cell signaling circuitry induce Ca²⁺ mobilization and signaling cascades that ultimately drive B cell proliferation, cytokine production and differentiation of naïve B cells and classical MBCs into ASCs (*Baba and Kurosaki, 2011*). Importantly, a variety of B cell inhibitory receptors can block BCR signaling and downstream effector functions (*Tsubata, 2012*) and members of the FCRL family, including FcRL3 and 5, can both enhance and inhibit B cell activation (*Ehrhardt and Cooper, 2011*; *Li et al., 2014*).

On the basis of the microarray data above, we tested the hypothesis that BCR signaling is attenuated in atypical MBCs relative to naïve B cells and classical MBCs. We mimicked BCR-antigen engagement in vitro by cross-linking the BCR with F(ab')2 IgG- and IgM-specific Abs for 5 min and compared the resulting phosphorylation of key downstream signaling molecules in naïve B cells, classical and atypical MBCs. Consistent with our hypothesis, we observed reduced phosphorylation of Syk, BLNK, PI3K, and PLCγ2 in atypical MBCs following BCR crosslinking compared to naïve B cells

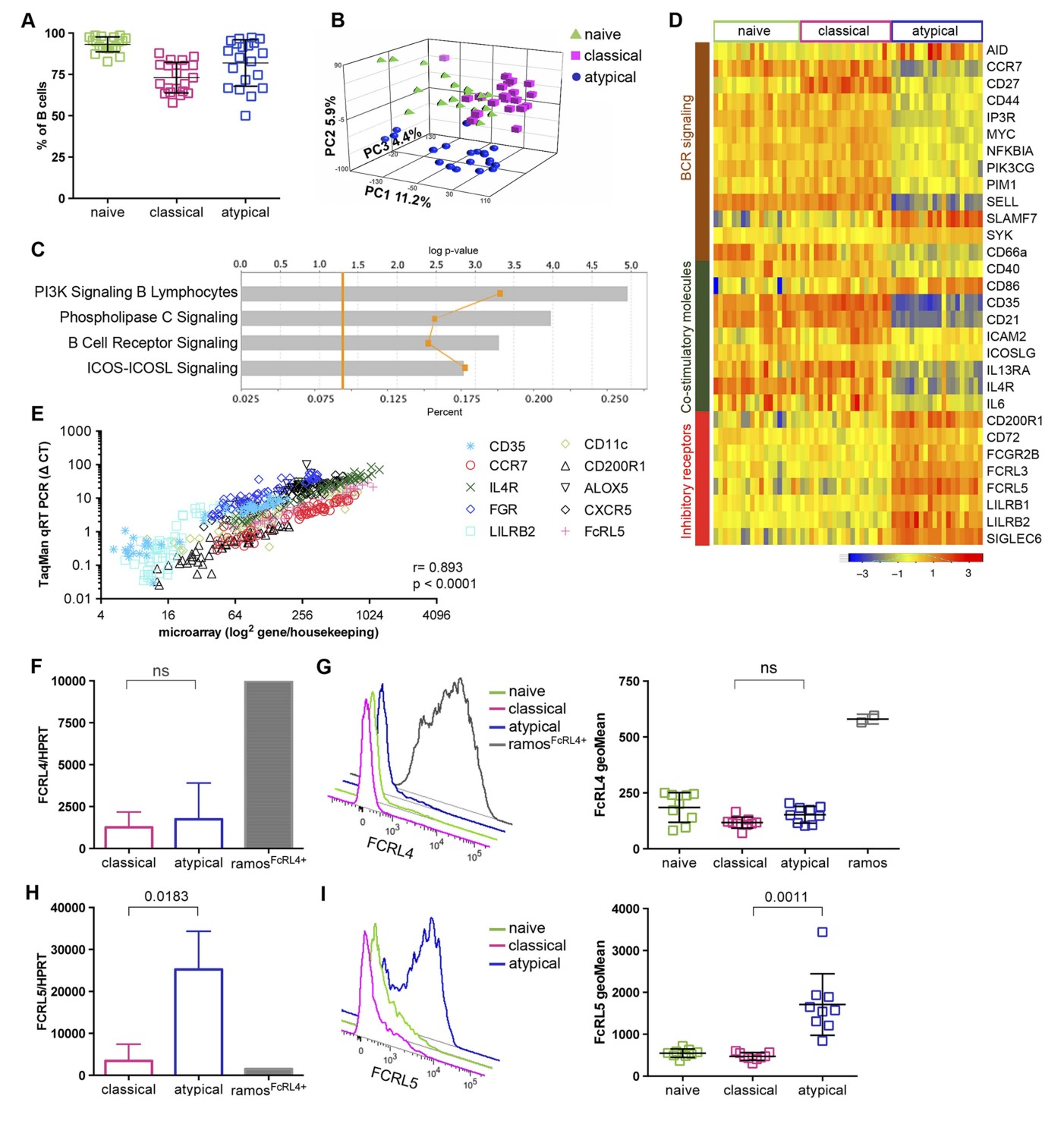

Figure 3. Atypical MBCs upregulate inhibitory receptors and differentially express genes involved in BCR and co-stimulatory signaling. (A) Naïve B cells (CD19+CD21+CD27−), classical (CD19+CD21+CD27+) and atypical MBCs (CD19+CD21−CD27−) as a percentage of total PBMCs after magnetic fractionation. Fractionation results of 20 Malian adults representative of subpopulations used in B–I. (B) Principal components analysis of gene expression microarray data. (C) IPA summary showing canonical pathways that distinguished atypical and classical MBCs. Yellow line indicates p values of the enrichment of each pathway. Vertical yellow line indicates threshold p value equivalent to 0.05. Gray bars indicate percentage of genes in each pathway that are differentially expressed among total number of genes in each pathway. (D) Heatmap showing ex vivo RMA-normalized log2 values of differentially expressed genes (DEGs) (rows) for each subject (columns) within each B cell subpopulation (n = 20 subjects). (E) qRT-PCR confirmation of microarray data (n = 20 subjects). (F) FCRL4 mRNA levels measured by qRT-PCR in B cell subpopulations and RAMOS-FcRL4+ cells (n = 4 subjects). (G) FcRL4 protein

*Figure 3. continued on next page*

Figure 3. Continued

expression measured by flow cytometry on B cell subpopulations and RAMOS-FcRL4+ cells; representative subject (left) and n = 10 subjects (right). (**H**) FCRL5 mRNA levels measured by qRT-PCR in B cell subpopulations and RAMOS-FcRL4+ cells (n = 4 subjects). (**I**) FcRL5 protein expression measured by flow cytometry on B cell subpopulations; representative subject (left) and n = 9 subjects (right). Horizontal bars and whiskers represent means and SD, respectively. p values determined by Spearman's (**E**), paired ttest (**F** and **H**) or ANOVA with Tukey's multiple comparisons test (**G** and **I**).
The following figure supplement is available for figure 3:

**Figure supplement 1**. Flow cytometry gating strategy to determine purity of naïve B cells (CD19+CD21+CD27−) (**A**), classical MBCs (CD19+CD21+CD27+) (**B**), and atypical MBCs (CD19+CD21−CD27−) (**C**) after magnetic sorting.

and classical MBCs (*Figure 4A* and *Figure 4—figure supplement 1*). Similarly, BCR crosslinking induced robust $Ca^{2+}$ mobilization in classical MBCs while $Ca^{2+}$ influx was markedly reduced in atypical MBCs, a finding confirmed by fluorescence microscopy (*Figure 4B*) and flow cytometry (not shown).

Together these results indicated that early BCR signaling events and $Ca^{2+}$ mobilization are diminished in atypical MBCs, and thus suggested that atypical MBCs may have reduced capacity to proliferate, produce cytokines and differentiate into ASCs in response to BCR crosslinking. Indeed, upon BCR crosslinking and stimulation with CD40-specific Abs, IL-10, IL-4 and CpG for 96 hr, proliferation of atypical MBCs was reduced compared to naïve B cells and classical MBCs (*Figure 4C*). Moreover, in response to BCR crosslinking and stimulation with CD40-specific Abs plus CpG, atypical MBCs failed to produce IL-6 and IL-8 (*Figure 4D*). Although TLR4 is expressed at low levels on resting human B cells, it is upregulated on B cells in individuals with inflammatory conditions (*Shin et al., 2009*; *Ganley-Leal et al., 2010*; *McDonnell et al., 2011*), thus we also attempted to activate atypical MBCs with LPS. Following a 12 hr stimulation with LPS alone, atypical MBCs did not produce CCL4 or IL-8 (*Figure 4E*).

Thus, malaria-associated atypical MBCs exhibit attenuated BCR signaling and $Ca^{2+}$ mobilization that manifests as reduced proliferative capacity and cytokine production.

## FcRL5 expression on atypical MBCs is associated with markedly reduced BCR signaling

As noted above, the FcRL5 protein was upregulated on atypical MBCs and its expression was bimodal (*Figure 3I*). To determine whether FcRL5 expression is associated with differential attenuation of BCR signaling in atypical MBCs, we crosslinked the BCR of FcLR5+ and FcRL5− atypical MBCs with F(ab')2 IgG- and IgM-specific Abs and compared the resulting phosphorylation of downstream signaling molecules. We observed that in FcRL5+ atypical MBCs the phosphorylation of PLCγ2 and to a lesser extent Syk was reduced as compared to FcRL5− atypical MBCs (*Figure 5A*). However, even FcRL5− atypical MBCs showed diminished phosphorylation of PLCγ2 relative to classical MBCs (all of which were FcRL5−) (*Figure 5A*), suggesting that a stepwise upregulation of inhibitory receptors leads to increasingly diminished BCR signaling, reminiscent of T cell exhaustion during persistent viral infections (*Kahan et al., 2015*). Consistent with this notion, we found that a higher percentage of FcRL5+ atypical MBCs co-expressed the inhibitory receptor FcγRIIb (*Figure 5B*).

## Atypical MBCs do not actively secrete Abs or differentiate into ASCs when stimulated

A recent study of *P. falciparum*-infected adults in Gabon suggested that atypical MBCs actively secrete Abs, on the basis of detecting secretory IgG transcripts in atypical MBCs (*Muellenbeck et al., 2013*). To address this possibility directly, we cultured magnetically sorted naïve B cells, classical and atypical MBCs for 48 hr without stimulation and then compared the frequency of ASCs by an IgM/IgA/IgG ELISPOT assay. As expected, naïve B cells and classical MBCs did not actively secrete Abs, while total PBMCs, which include plasmablasts and plasma cells, contained a low frequency of ASCs as expected (*Figure 6A*). However, we found that atypical MBCs showed no evidence of active Ab secretion (*Figure 6A*).

Given the important implications of this finding for interpreting the role of atypical MBCs in humoral immunity, we sought to verify that the sorting conditions of the previous experiment did not

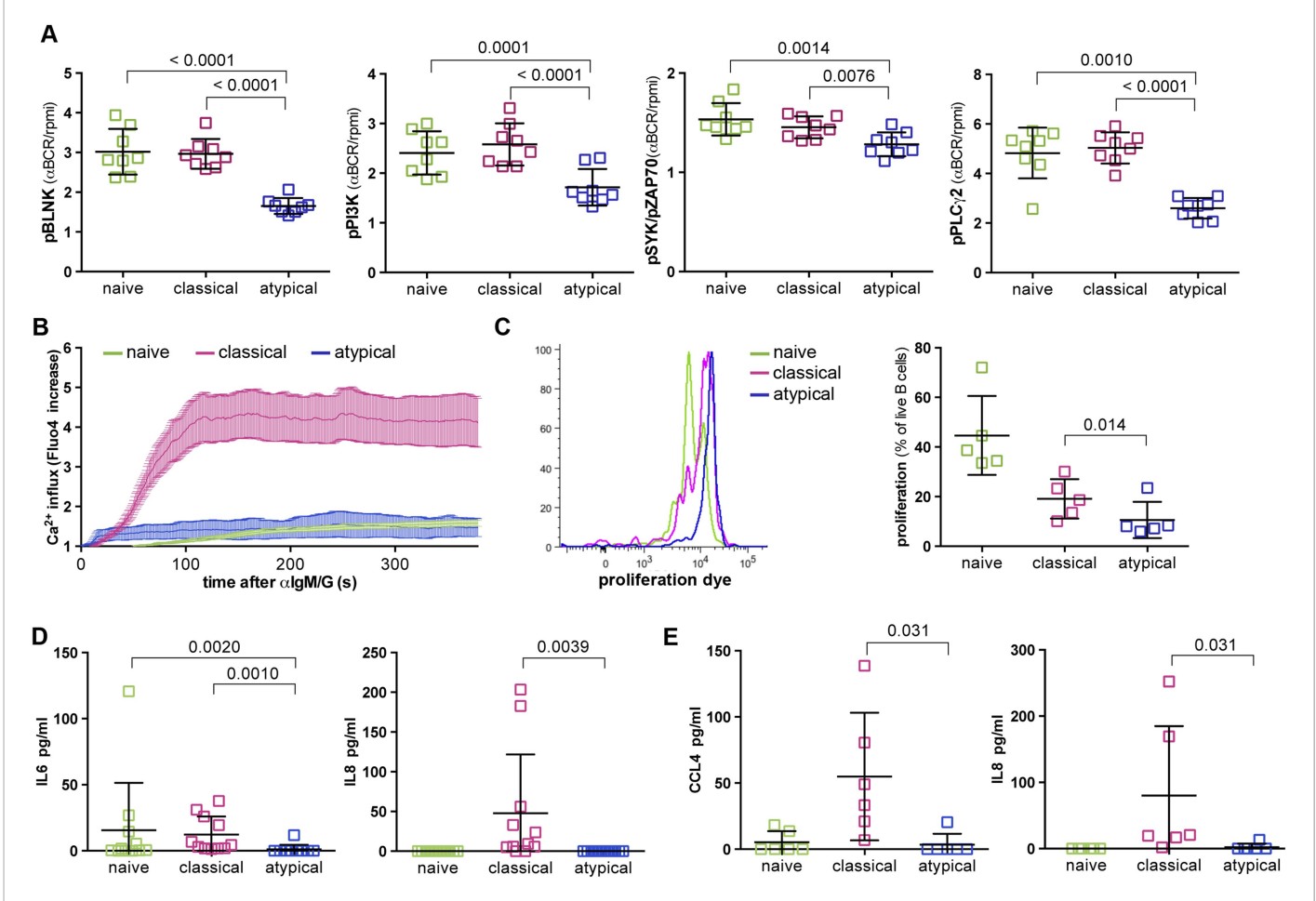

**Figure 4**. Atypical MBCs exhibit reduced BCR signaling, Ca²⁺ mobilization, proliferation and cytokine production. (**A**) Following 5 min of BCR cross-linking with anti-IgM /G Abs, magnetically sorted naïve B cells (CD19⁺CD21⁺CD27⁻), classical MBCs (CD19⁺CD21⁺CD27⁺) and atypical MBCs (CD19⁺CD21⁻CD27⁻) were incubated with Abs specific for phosphorylated BLNK, PI3K, Syk and PLCγ2 and analyzed by flow cytometry. Shown are ratios of phosphorylation induced by BCR cross-linking over tonic signal fluorescence in RPMI (n = 8 subjects). (**B**) Flow cytometry analysis of Ca²⁺ influx in B cell subpopulations following BCR-crosslinking. Plot representative of 4 independent experiments. (**C**) Proliferation of B cell subpopulations assessed by flow cytometry following stimulation with anti-IgM/anti-IgG, anti-CD40, IL-10, IL-4 and CPG for 96 hr. Representative MFI histogram shown on left. (**D**) IL-6 and IL-8 production by B cell subpopulations measured in supernatants following 12 hr stimulation with anti-IgM/anti-IgG, anti-CD40 and CPG (n = 6 subjects). (**E**) CCL4 and IL-8 production by B cell subpopulations measured in supernatants following 12 hr stimulation with LPS (n = 6 subjects). Horizontal bars and whiskers represent means and SD, respectively. p values determined by ANOVA with Tukey's multiple comparisons test (**A**, **D** and **E**) or paired t test (**C**).

The following figure supplement is available for figure 4:

**Figure supplement 1**. Histogram of phosphorylated BLNK (**A**), phosphorylated SYK (**B**), phosphorylated PI3K (**C**) and phosphorylated PLCγ2 (**D**) following 5 min of BCR cross-linking with anti-IgM/G Abs (red) or tonic signal in RPMI (salmon) of magnetically sorted naïve B cells (CD10⁻CD19⁺CD20⁺ CD21⁺CD27⁻), classical MBCs (CD10⁻CD19⁺CD20⁺CD21⁺CD27⁺) and atypical MBCs (CD10⁻CD19⁺CD20⁺CD21⁻CD27⁻) of a representative Malian adult analyzed by flow cytometry.

inhibit Ab secretion from atypical MBCs. To do so, we fractionated PBMC samples using two different negative B cell selection cocktails that only differed by the presence of CD43- and CD66b-specific Abs—proteins that are expressed exclusively on plasmablasts and plasma cells. This yielded from each sample two fractions: total B cells, and total B cells without plasma cells and plasmablasts (***Figure 6B, C***). As expected, the B cell fraction that included plasma cells and plasmablasts contained ASCs (***Figure 6D***), whereas the fraction from which plasma cells and plasmablasts had been depleted did not contain ASCs (***Figure 6D***), thus confirming that atypical MBCs do not actively secrete Abs.

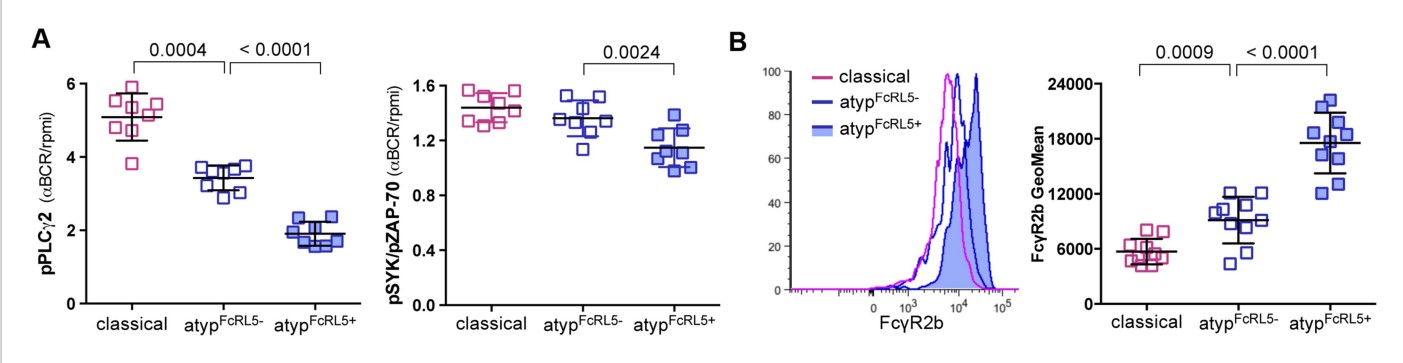

**Figure 5.** FcRL5 expression on atypical MBCs is associated with markedly reduced BCR signaling. (**A**) Following 5 min of BCR cross-linking with anti-IgM and anti-IgG Abs, PBMCs were incubated with Abs specific for phosphorylated PLCγ2 (left) and Syk (right). Classical MBCs, and FcRL5⁻ and FcRL5⁺ atypical MBCs were analyzed by flow cytometry. Shown are ratios of phosphorylation induced by BCR cross-linking over tonic signal fluorescence in RPMI (n = 7 subjects). (**B**) MFI of FcγR2b expression by flow cytometry on classical MBCs, and FcRL5⁻ and FcRL5⁺ atypical MBCs (n = 10 subjects). Representative MFI histogram shown on left. Horizontal bars and whiskers represent means and SD, respectively. p values determined by ANOVA with Tukey's multiple comparisons test (**A** and **B**).

Because the recent study in Gabon analyzed atypical MBCs from adults with acute febrile malaria (*Muellenbeck et al., 2013*), it remained possible that atypical MBCs only secrete Abs during an active malaria infection. To test this hypothesis, we used the same negative selection procedures described above to isolate total B cells and B cells without plasma cells/plasmablasts from children during acute febrile malaria. With these fractions we found by ELISPOT that atypical MBCs do not actively secrete Abs during an active malaria infections (*Figure 6E*).

Finally, we asked whether atypical MBCs could differentiate into ASCs in vitro when stimulated with CpG oligonucleotide, Protein A from *Staphylococcus aureus*, pokeweed mitogen and IL-10 (*Weiss et al., 2012*). In contrast to classical MBCs, which readily differentiate into ASCs in response to these polyclonal stimuli, atypical MBCs did not differentiate into ASCs (*Figure 6F*).

Thus, malaria-associated atypical MBCs do not actively secrete Abs at rest or in response to a malaria infection in vivo. Moreover, atypical MBCs do not differentiate into ASCs when stimulated in vitro. Together these data indicate that atypical MBCs do not directly contribute to Ab-mediated protection from malaria.

## Discussion

Two features that distinguish the humoral immune response to *P. falciparum* infection in individuals living in malaria endemic areas are the inefficiency with which long-lived plasma cells and classical MBCs are acquired and the large expansion of atypical MBCs, even in young children (*Portugal et al., 2013*; *Scholzen and Sauerwein, 2013*; *Hviid et al., 2015*). As a consequence of the slow acquisition of malaria-specific MBCs and long-lived Abs, children in endemic areas are left susceptible to repeated bouts of clinical malaria that in about 1% of cases becomes severe and life threatening (*WHO, 2013*). Although the expansion of atypical MBCs appears to be driven by malaria (*Weiss et al., 2011*; *Illingworth et al., 2013*; *Muellenbeck et al., 2013*; *Scholzen et al., 2014*), we do not yet know the consequences of this expansion.

To gain insight into the developmental relationship between atypical and classical MBCs we carried out next generation sequencing of the expressed $V_H$ genes of each subpopulation and also determined the number of cell divisions cells that each subpopulation had undergone. Indeed, $V_H$ gene sequencing showed that $V_H$ gene usage and the rates of SHM in atypical and classical MBCs, predicted to occur in germinal center reactions, were indistinguishable. Notably, we observed a high degree of clonal connectivity between atypical and classical MBCs, indicating that a significant proportion of atypical and classical MBCs share a common precursor. By KREC analysis we also observed that atypical and classical MBCs appear to have undergone a similar number of cell divisions. Together the results of these analyses suggest that atypical and classical MBCs are closely related developmentally and have had similar antigen-driven germinal center experiences.

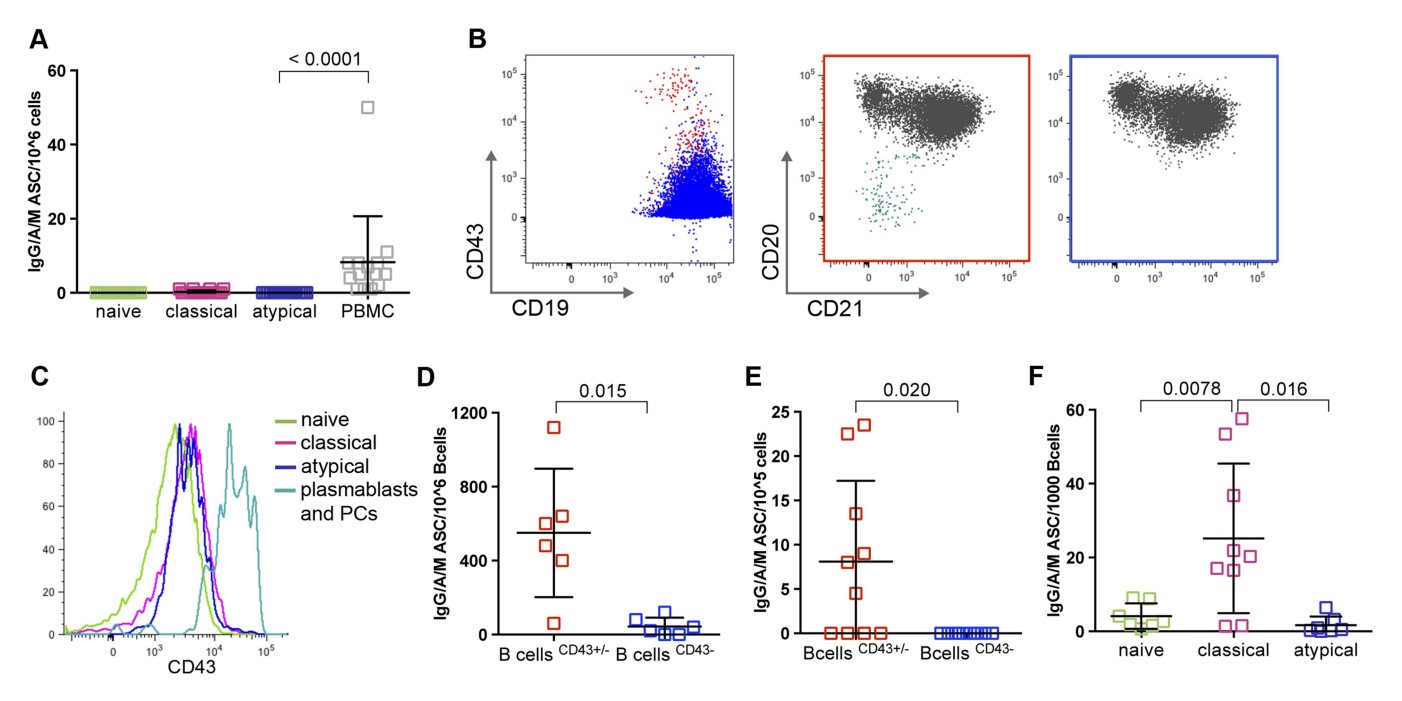

**Figure 6**. Atypical MBCs do not actively secrete Abs or differentiate into ASCs. (**A**) Total PBMCs and magnetically sorted naïve B cells (CD19+CD21+CD27−), classical (CD19+CD21+CD27+) and atypical MBCs (CD19+CD21−CD27−) were cultured for 48 hr without stimulation, then ASCs were quantified by IgM/IgA/IgG ELISPOT (n = 14 subjects). (**B**) Magnetic sorting strategy for experiments described in **D** and **E**. B cells without plasma cells/plasmablasts were negatively selected with Abs to CD2, CD3, CD14, CD16, CD36, CD43, CD56, CD66b, and glycophorin A (left panel, blue population), while total B cells were negatively selected with Abs to CD2, CD3, CD14, CD16, CD56, and glycophorin A (left panel, red population). Red population contains plasma cells/plasmablasts (CD19+CD20−CD21−) (middle panel, aqua population) whereas blue population does not (right panel). (**C**) Representative histogram showing specificity of CD43 for plasma cells/plasmablasts. (**D**) Total B cells (CD43+/− fraction) and B cells without plasma cells/plasmablasts (CD43− fraction) were isolated from Malian adults and cultured for 48 hr without stimulation, then ASCs were quantified by IgM/IgA/IgG ELISPOT (n = 6 subjects). (**E**) Total B cells and B cells without plasma cells/plasmablasts were isolated from children during acute malaria and cultured for 48 hr without stimulation, then ASCs were quantified by IgM/IgA/IgG ELISPOT (n = 10 subjects). (**F**) Magnetically sorted B cell subpopulations were cultured for 6 days with CpG, IL-10, SAC and PWM and ASCs were quantified by IgM/IgA/IgG ELISPOT (n = 9 subjects). Horizontal bars and whiskers represent means and SD, respectively. p values determined by ANOVA with Tukey's multiple comparisons test (**A** and **F**) or paired ttest (**D** and **E**).

To obtain a more in-depth view of the similarities and differences between atypical and classical MBCs we analyzed and compared the transcriptional profiles of the two populations. This analysis revealed the differential expression of several inhibitory receptors including ones that we previously determined were upregulated in atypical MBCs by cell surface phenotyping (*Weiss et al., 2009*). Two DEGs of particular interest are FCRL5 and FCRL3. By cell surface phenotyping we originally reported that atypical MBCs expressed FcRL4 (*Weiss et al., 2009*). The FcRL4 expression was low and because of the absence of FCRL4 transcripts in atypical MBCs reported here we now conclude that the staining of B cells with FcRL4-specific mAb was likely due to cross reactivity of the mAb with FcRL5 or FcRL3. The best-characterized B cell populations that resemble atypical MBCs, namely tissue–like MBCs described in the mucosal lymphoid tissues of healthy individuals (*Ehrhardt et al., 2005*) and exhausted MBCs described in HIV-infected individuals (*Moir et al., 2008*), express FcRL4 at both the protein and transcript levels. These results suggest the interesting possibility that the differential expression of FCRLs by B cell populations may be dictated by the immune environment in which the cells arise.

How FcRL5 and FcRL3 contribute to the biology of atypical MBCs remains to be determined. In mice, FcRL5 expression is associated with innate-like B cell lineages (*Li et al., 2014*) and recent studies suggest that FcRL5, which contains both an immunoreceptor tyrosine-based inhibition motif (ITIM) and immunoreceptor tyrosine-based activation motif (ITAM) within its cytoplasmic domain, may exert a binary, compartment-specific influence on B cell responses (*Zhu et al., 2013*). In humans, FcRL5 expression has been linked to both inhibition and enhancement of B cell responses. FcRL5, through its

recruitment of the phosphatases SHP1 and SHP2, inhibits BCR-dependent activation of human B cells (*Haga et al., 2007*). In human naïve B cells, BCR crosslinking was shown to induce the expression of FcRL5 and crosslinking FcRL5 together with TLR9 activation enhanced B cell proliferation (*Dement-Brown et al., 2012*). FcRL5 has been recently shown to bind to Fc of Ig (*Wilson et al., 2012*) and would be predicted to respond to environmental changes in the levels of immune complexes. Intriguingly, FcRL5 has also been shown to bind to an orthopoxvirus MHC-class I-like molecule secreted by infected cells suggesting a role of FcRL5-expressing cells in defense against poxvirus infection (*Campbell et al., 2010*). It will be interesting to explore whether FcRL5 binds to any *Plasmodium*-encoded products. The human FcRL3 also contains both ITIMs and ITAMs in its cytoplasmic domain and appears to be most highly related to FcRL5 in mice (*Li et al., 2014*). Recent studies of human peripheral blood B cells provided evidence that FcRL3 expression increases as a function of differentiation and is highest among MBC subsets with innate-like features (*Li et al., 2013*). FcRL3 ligation augmented TLR9 mediated B cell proliferation but blocked plasma cell differentiation and Ab production. Thus, it is possible that both FcRL5 and FcRL3 serve to orchestrate adaptive and innate responses in atypical MBCs.

Importantly, we provide evidence here that atypical MBCs have lost two key adaptive immune B cell functions, namely the ability to signal through the BCR and to differentiate into ASCs. Attenuated BCR signaling was suggested by the microarray data that showed differential expression of key components of the BCR signaling pathway. A direct test of the ability of atypical MBCs to respond to BCR crosslinking showed that signaling was blocked at an early signaling event, the phosphorylation of the kinase Syk. It is reasonable to assume that this block is mediated by the several inhibitory receptors upregulated by atypical MBCs, including FcRL5, through their ability to recruit the phosphatases SHP1/SHP2 (*Li et al., 2014*). The inability of atypical MBCs to respond to antigen stimulation raises the question: to what, if anything, are atypical MBCs B cells sensitive? We recently demonstrated that the expression of FcRL4 similarly blocked BCR signaling but simultaneously enhanced responses to TLR9 ligands and we suggested that FcRL4 expression switched adaptive B cells to innate-like B cells (*Sohn et al., 2011*). Although here we observed that atypical MBCs showed reduced responsiveness to TLR4 and TLR9 ligands and other polyclonal stimuli, the extent to which this is modulated by FcRL5 expression remains to be determined.

We also demonstrate that atypical MBCs do not spontaneously secrete Abs and moreover cannot be stimulated to secrete Abs under condition that induce classical MBCs to readily differentiate into ASCs. We observed a lack of spontaneous Ab secretion from both 'resting' atypical MBCs in Malian adults as well as from atypical MBCs isolated from children during acute febrile malaria. These results indicate that atypical MBCs do not directly contribute to Ab-mediated protection from malaria. Our findings are in contrast to a recent study of *P. falciparum*-infected adults living in Gabon in which it was suggested that atypical MBCs actively secrete Abs, an assertion based on the expression of mRNA encoding secretory Ig, although Ig secretion was not measured directly (*Muellenbeck et al., 2013*). It is possible that under some conditions atypical MBCs may secrete Ig but what those conditions are remains to be determined. It will be of interest in future longitudinal studies to compare the biochemical and molecular features of atypical and classical MBCs within individuals before, during and after *P. falciparum* infections and clinical malaria.

Profiling the antigen specificity of atypical MBCs will ultimately be necessary to fully understand the biology of these cells. Because atypical MBCs do not actively secrete Abs or differentiate into ASCs when stimulated, we could not directly assess their antigen specificity through conventional methods such as ELISA or ELISPOT. In ongoing efforts we are attempting to develop fluorescently labeled *P. falciparum* antigens to identify antigen-specific B cells by flow cytometry, although this approach is challenging due to the low frequency of any given antigen-specific MBC population in peripheral blood. Because unambiguous targets of protective Abs have yet to be identified in malaria, another approach involves cloning and expressing antibodies from bulk-sorted atypical and classical MBCs to compare their specificity profiles in an unbiased manner using protein microarrays that represent the entire *P. falciparum* proteome of over 5400 gene products.

Whether atypical MBCs negatively impact the efficacy of vaccines in malaria endemic areas, particularly candidate malaria vaccines, also remains to be determined. This possibility was suggested by the observation that the TLR9 agonist CpG failed to enhance specific MBC responses to a candidate malaria vaccine in Malian adults (*Traore et al., 2009*) but not U.S. adults (*Crompton et al., 2009*). It will be of interest to determine whether reduced BCR signaling and effector function in

atypical MBCs can be reversed in vitro, and if so, whether this can be translated into strategies that enhance vaccine efficacy in malaria-exposed populations. Further insight into whether atypical MBCs negatively impact vaccine-induced or naturally acquired immunity to malaria and other pathogens would be facilitated by reagents that reliably identify antigen-specific B cells.

In summary, while atypical and classical MBCs appear closely related developmentally, atypical MBCs exhibit markedly reduced signaling and effector function, which may contribute to the inefficient acquisition of humoral immunity to malaria. These findings provide fundamental insights into the functional plasticity of the human B cell compartment, and also suggest that B cells could be modulated to enhance vaccine efficacy, particularly in malaria endemic areas.

## Materials and methods

### Study subjects

#### Malian donors
For this study we focused on asymptomatic healthy adults (n = 107) aged 18–61 years with lifelong malaria exposure. These adults, for whom asymptomatic blood-stage *P. falciparum* infection is common (*Crompton et al., 2008*), were enrolled in NIAID protocols 07-I-N141 or 06-I-N147, both of which were conducted in Kambila, Mali, a rural village of ~1500 inhabitants where intense *P. falciparum* transmission occurs from July through December each year (*Crompton et al., 2008*). We also included 6 children enrolled in protocol 06-I-N147 who had PBMCs collected during an acute febrile malaria episode. Enrollment exclusion criteria for both protocols included anemia (hemoglobin <11 g/dl); current use of antimalarials, corticosteroids or other immunosuppressants; fever >37.5°C or evidence of an acute infection; and current pregnancy. After enrollment, the same exclusion criteria applied to all scheduled peripheral blood draws (either 16 or 48 ml for adults depending on the protocol, or 8 ml for children), which occurred at multiple cross-sectional time points between May 2006 and May 2014.

#### U.S. donors
Peripheral blood samples from 37 healthy U.S. adult donors enrolled in NIH protocol # 99-CC-0168) were also analyzed. Demographic and travel history data were not available from these anonymous donors, but prior *P. falciparum* exposure is unlikely.

### Sample processing
In mali, blood was collected by venipuncture into sodium citrate-containing cell preparation tubes (Vacutainer CPT Tubes BD, Franklin Lakes, NJ) and transported 20 km to the laboratory where PBMCs were isolated according to the manufacturer's instructions. PBMCs were analyzed immediately or frozen within 3 hr for later analysis. PBMCs were frozen in fetal bovine serum (FBS) (Gibco, Grand Island, NY) containing 7.5% dimethyl sulfoxide (DMSO; Sigma–Aldrich, St. Louis, MO), kept at −80°C for 24 hr, and then stored at −196°C in liquid nitrogen. U.S. blood samples were drawn into heparinized tubes (BD) and PBMCs were isolated from whole blood by Ficoll-Hypaque density gradient centrifugation (GE Healthcare, Uppsala, Sweden) according to the manufacturer's instructions, and frozen under the same conditions as the Malian samples. For all assays, PBMCs were rapidly thawed in a 37°C water bath, washed in PBS with 10% heat-inactivated FBS and then in complete RPMI (RPMI 1640 with L-glutamine supplemented with 10% heat-inactivated FBS, penicillin/streptomycin 10,000 μg/ml, and 50 μM β-Mercaptoethanol [all from GIBCO, Invitrogen]). For each experiment, PBMCs from all individuals were thawed and assayed at the same time. The trypan blue dye exclusion assay consistently demonstrated >80% viability of PBMCs after thawing.

### B cell phenotypic analysis
One million PBMCs were washed in PBS with 4% heat-inactivated FBS and incubated for 30 min at 4°C with mouse monoclonal fluorescently labelled Abs specific for surface molecules. *Supplementary file 3* provides the full list of Abs used. Flow cytometry was performed with a BD LSR II Table flow cytometer (BD Biosciences) and data were analysed using FlowJo software (Tree Star, Inc, Ashland, OR).

### B cell FACS sorting
Immediately after thawing 4–10 pooled vials of PBMCs, each containing ~$10^6$ cells from the same donor, B cells were isolated with the negative selection Human B Cell Enrichment Kit (STEMCELL

Technologies, Vancouver, Canada) according to the manufacturer's instructions. Purity of enriched B cells was consistently >98% with an average of 98.7% (95% CI: 97.6–99.9). The number of isolated B cells ranged from 3 to 8 million *per* donor. These were stained with anti- CD10 APC (CB-CALLA), CD19 PERCP-Cy5.5 (SJ25C1) (eBioscience, San Diego, CA), CD20 APC-H7 (L27) and CD21 PE-Cy5 (B-ly4) (BD Biosciences); and CD27 BV421 (M-T271) (BioLegend, San Diego, CA) conjugated Abs. Lymphocytes were gated using a FSC vs SSC plot following doublet discrimination. The singlet lymphocytes were then gated on $CD19^+CD20^+$ cells followed by a $CD10^-$ gate to identify mature B cells. Further discrimination using CD21 and CD27 allowed gating on atypical MBCs ($CD10^-CD19^+CD20^+CD21^-CD27^-$), classical MBCs ($CD10^-CD19^+CD20^+CD21^+CD27^+$) and naïve B cells ($CD10^-CD19^+CD20^+CD21^+CD27^-$) that were sorted using a FACSAria (BD Biosciences). This method produced $3 \times 10^5$ to $10^6$ highly pure naïve B cells, classical and atypical MBCs.

## B cell magnetic fractionation

Immediately after isolating fresh PBMCs from 48 ml of blood or thawing 1 to 3 frozen vials of PBMCs from the same donor obtained form 16 ml of blood each, B cells were isolated with the negative selection Human B Cell Enrichment Kit (STEMCELL Technologies) according to the manufacturer's instructions. Following enrichment, B cells were fractionated into atypical MBCs ($CD19^+CD21^-CD27^-$), classical MBCs ($CD19^+CD27^+CD21^+$) and naïve B cells ($CD19^+CD21^+CD27^-$) using a three-step magnetic bead-based selection procedure. B cells were incubated with PE-conjugated anti-CD21 mAB (HB5) (eBioscience), followed by Anti-PE MultiSort MicroBeads (Miltenyi Biotec, Bergisch Gladbach, Germany) according to the manufacturer's instructions to separate $CD21^-$ and $CD21^+$ B cells. Following positive selection, the magnetic particles were removed from the $CD21^+$ fraction using the MultiSort Release Reagent according to the manufacturer's instructions. Finally, $CD21^-$ and $CD21^+$ B cells were incubated separately with anti-CD27 MicroBeads (Miltenyi Biotec) according to the manufacturer's instruction to separate $CD27^-$ and $CD27^+$ B cells. The purity of each fraction was verified by flow cytometry using fluorescently labeled Abs specific for CD19 PerCP-Cy5.5 (SJ25C1), CD21 PE (HB5) (eBioscience) and CD27 APC (O323) (invitrogen, Life Technologies, Carlsbad, CA). This method produced $10^5$ to $10^6$ highly pure naïve B cells, classical and atypical MBCs depending on the starting volume of blood.

## Next generation sequencing of the IgH repertoire

Total cellular RNA was isolated from $\sim 2 \times 10^5$ sorted naïve B cells, classical and atypical MBCs using the RNeasy Micro kit by following the manufacturer's protocol (Qiagen, Valencia, CA). Approximately 10 ng of RNA was subjected to reverse transcription using the iScript cDNA synthesis kit (BioRad, Hercules, CA). Aliquots of the resulting single-stranded cDNA products were mixed with 50 nM of VH1-VH7 FR1 specific primers and 250 nM Cα and Cγ or Cμ specific primers preceded by the respective Illumina nextera sequencing tag (sequences listed below) in a 25 μl PCR reaction (using 4 ml template cDNA) using Invitrogen's High Fidelity Platinum PCR Supermix (Invitrogen). Amplification was performed with a Bio-Rad C1000 Thermal Cycler (Bio-Rad) with the following conditions: after an initial step of 95°C for 5 min, 25–40 cycles (depending on amplification efficiency) of 95°C for 30 s, 55°C for 30 s, and 72°C for 30 s, ending with a final extension step of 72°C for 5 min. Nextera indices were added via PCR with the following conditions: 72°C for 3 min, 98°C for 30 s, 5 cycles of 98°C for 10 s, 63°C for 30 s, and 72°C for 3 min. Ampure XP beads (Beckman Coulter Genomics, Danvers, MA) were used to purify the products and they were subsequently pooled and denatured. Single strand products were sequenced on a MiSeq (Illumina, San Diego, CA) using the 300bp x2 v3 kit. VH1a: CAGGTKCAGCTGGTGCAG, VH1b: SAGGTCCAGCTGGTACAG, VH1c: CARATGCAGCTGGTGCAG, VH2: CAGGTCACCTTGARGGAG, VH3: GGTCCCTGAGACTCTCCTGT, VH4: ACCCTGTCCCTCACCTGC, VH5: GCAGCTGGTGCAGTCTGGAG, VH6: CAGGACTGGTGAAGCCCTCG, VH7: CAGGTGCAGCTGG TGCAA) Cm: CAGGAGACGAGGGGGAAAAGGCg: CCGATGGGCCCTTGGTGGA, Ca: GAAGACCTTG GGGCTGGTCG) F sequencing tag: TCGTCGGCAGCGTCAGATGTGTATAAGAGACAG R sequencing tag: GTCTCGTGGGCTCGGAGATGTGTATAAGAGACAG

## Analysis of next generation sequencing data

An informatics pipeline (*Figure 2—figure supplement 2*) developed by Dr Alex Rosenberg and Chris Fucile at the University of Rochester School of Medicine was used for analysis of sequencing data (Tipton et al. in preparation). After paired-end reads were joined, sequences were filtered based on

a length and quality threshold. Sequences less than 200 bp and sequences with poor overlaps (>8% difference in linked region) and/or high number of bp below a threshold score (sequences containing more than 15 bp with less than Q30, 10bp with less than Q20, or any bp with less than Q10 scores) were excluded from further analysis. Isotypes were then determined by analysis of the constant region segment of each sequence and a random subset of 100,000 sequences were aligned and analyzed for clonality and for mutations in the V region using the data provided by IMGT/HighV-quest (http://www.imgt.org/HighV-QUEST/). The subset of sequences was used to relieve computational stress and allow for analysis in reasonable timing. Samples tested at multiple sized subsets did not result in any substantial difference in the clonality or our interpretation of results. After alignment, the sequences were analyzed by a custom program written in perl and matlab. Sequences were filtered by removing 'unproductive' and 'unknown' sequences, and clusters of sequences were identified based on identical V and J region rearrangement, identical HCDR3 length, and 85% similarity within the HCDR3. All of IMGT/HighV-quest's data is retained through the process and used for mutation calculations. The frequency and distribution of SHM were ascertained on the basis of non-gap mismatches of expressed sequences with the closest germline VH sequence.

## RNA extraction and microarray target preparation

For the microarray analysis, RNA extraction, cDNA synthesis, amplification and labeling were performed as described previously, with some modifications (*Li et al., 2014*). Briefly, $5 \times 10^5$ to $10^6$ isolated naïve B cells, classical and atypical MBCs B cells were placed into 500 μl RLT lysis buffer (Qiagen, Valencia, CA) containing 0.145 M β-mercaptoethanol. The sample was passed through a Qiashredder column (Qiagen) at 21,000×$g$ for 2 min to shear high molecular weight gDNA and increase RNA yields. Next, an equal volume of 70% ethanol was added and samples were placed into a pre-assigned well of a randomized and balanced 96-well format to reduce confounding and batch effects due to plate position. Randomization and balancing was performed based upon B cell population, subject ID, and phlebotomy date and no samples were placed in the outside wells in an effort to avoid edge effects due to evaporation. RNAs were purified using the RNeasy 96 Kit following manufacturer's instructions, including the on-column DNase treatment (Qiagen). Purified RNAs were quantified using the Quant-iT RiboGreen RNA fluoremetric assay (Life Technologies). RNA purity (A260/A280) and quality were assessed using UV spectrophotometry and Agilent's 2100 Bioanalyzer (Agilent Technologies, Santa Clara, CA), respectively. Ten nanograms of each RNA was vacuum concentrated to 5 μl and cDNA was synthesized using the Ovation Pico WTA System v2 (Nugen, San Carlos, CA). SPIA-amplified cDNAs were purified using Agencourt RNAClean XP magnetic beads (Beckman Coulter, Indianapolis, IN) and analyzed on the Agilent 2100 Bioanalyzer using the Pico analysis kit (Agilent). Five micrograms of each cDNA was fragmented and biotin-labeled using the Encore Biotin Module and instructions (Nugen).

## Microarray chip processing and data analysis

Hybridization, fluidics and scanning were performed according to standard Affymetrix protocols (http://www.affymetrix.com). Command Console (CC v3.1, http://www.affymetrix.com) software was used to convert the image files to cell intensity data (cel files). All cel files, representing individual samples, were normalized by using the robust multi-chip analysis (RMA) method within expression console (EC v1.3, http://www.affymetrix.com) to produce the analyzed cel files (chp files) along with the report files. The cel files were input into Partek Genomics Suite software (Partek, Inc. St. Louis, MO, v6.6-6.12.0907) and grouped to produce the principal components analysis (PCA) graph and dendrogram. An ANOVA was performed within Partek to obtain multiple test corrected p-values using the false discovery rate method at the 0.05 significance level and was combined with fold change values, signal confidence (above background), and call consistency (as a percent) calculated using custom Excel templates for each comparison of interest. The resulting data were analyzed using IPA (Ingenuity Pathway Systems, www.ingenuity.com) generating the networks, functional analyses and pathway construction. The array data discussed in this publication have been deposited in NCBI's Gene Expression Omnibus (Edgar et al., 2002) and are accessible through GEO series accession number GSE65928. (http://www.ncbi.nlm.nih.gov/geo/query/acc.cgi?acc=GSE65928).

## Quantitative real-time PCR

Constitutively expressed housekeeping gene candidates were selected from the DNA microarray results. Genes were ranked by probe-set expression level and by coefficient of variation (CV). Three genes were selected based on three criteria: uniform mRNA expression in all samples, low CV across treatments, expression level and gene function. The three genes were tRNA 5-methylaminomethyl-2-thiouridylate methyltransferase (TRMU), ubiquitination factor E4A (UBE4A), and lysine (K)-specific demethylase 5A (KMD5A), which were expressed from low to high level, respectively. All reference and validation gene TaqMan expression assays were ordered from Life Technologies. All TaqMan assays stated with 'Best Coverage' for RNA analysis by the company were selected (*Supplementary file 2C*). QPCR amplification efficiency was tested for each constitutively expressed gene in combination with each target gene. The Invitrogen Express QPCR supermix universal with premixed ROX (Life Technologies) was used to perform the assays. The multiplex reactions were carried out in 20 µl reactions with 1× RT-PCR buffer, 1× TaqMan expression assay mix. The Q-PCR reactions were carried out at 50°C for 2 min, 95°C for 2 min, and 55 cycles of 95°C for 15 s and 60°C for 1 min. Data was analyzed using 7900HT version 2.4 sequence detection system software according to manufacturer's recommendations (Life technologies). All three reference gene primers and probes performed at an efficiency of 85% or above in duplex QPCR reactions with the 10 target genes selected for validation, with the exceptions being IL4R and KMD5A. The UBE4A reference gene was selected due to its medium level of expression and consistent amplification efficiency across all 10 genes. 10 QPCR master mixes were prepared with each consisting of the target gene, UBE4A reference gene and TaqMan expression mix. Each mastermix was placed on a 96-well plate and ss cDNA sample template was added to each well in a 96-well plate. Each mastermix was combined with template cDNA on a single 384-well plate using Biomek NX$^P$ automated lab assistant (Beckman Coulter Inc.) to minimize well-to-well and plate-to-plate pipetting differences in cDNA template and master mix volumes. All QPCR reactions were prepared on the same day by one individual. Plates were frozen at −80°C, thawed one-by-one, and run on 7900HT TaqMan instrument, and analyzed following the comparative $C_T$-method (Life Technologies). Spearman non-parametric correlation (GraphPad Software, La Jolla, CA) was calculated between real-time PCR $C_T$-values and quantile normalized GeneChip probe-set signals for the 10 target genes for all 60 samples.

## Replication history of sorted B cell subpopulations

The Kappa-deleting Recombination Excision Circles (KREC) assay was performed as described previously (*van Zelm et al., 2007*). Briefly, coding and signal joints of the IGK-deleting rearrangement were quantified by qRT-PCR from genomic DNA of $3 \times 10^5$ to $10^6$ FACS sorted naïve B cells, classical and atypical MBCs on an ABI Prism 7500 (Applied Biosystems, Life Technologies). The number of cell divisions experienced by each cell population was determined by the ratio of the number of coding joints and signal joints.

## B cell stimulation for ELISPOT

Two to $5 \times 10^5$ magnetically sorted naïve B cells, classical and atypical MBCs were cultured in complete media in 24-well plates with polyclonal stimulants for MBC activation as previously described (*Weiss et al., 2012*). Briefly, cells were cultured for 6 days with 2.5 µg/ml of CpG oligodeoxynucleotide ODN-2006 (Operon Technologies, Huntsville, AL); 25 ng/ml of IL-10, (R&D Systems, McKinley Place, NE); 1/10,000 dilution of Protein A from *S. aureus* Cowan (SAC) and 1/100,000 dilution of pokeweed mitogen (PWM) (Sigma–Aldrich, St. Louis, MO).

## B cell ELISPOT

Directly after thawing or after the 6-day culture with CpG, IL-10, SAC and PWM (described above), cells were washed in complete media, warmed to 37°C, counted and plated on ELISPOT plates (Millipore Multiscreen MSIP HTS IP Sterile plate 0.45 µm, hydrophobic, high-protein binding, Millipore, Billerica, MA) pre-coated with 10 µg/ml polyclonal goat Abs specific for human IgG (Caltag, Life Technologies) and blocked with a solution of 1% BSA in RPMI. Serial dilutions of the different B cell subpopulations at concentrations ranging from $2 \times 10^4$ to 156 cells were then incubated at 37°C in a 5% $CO_2$ incubator for 6 hr, and then washed 4 times each with PBS and PBS-0.05% Tween 20 (PBST). Goat Abs specific for human IgG Fc or IgG, IgM, IgA (H+L) conjugated to alkaline phosphatase (Jackson ImmunoResearch Laboratories, West Grove, PA; and Thermo Scientific, Life

Technologies respectively) diluted 1:1000 or 1:5000, respectively, in PBS 0.05% Tween 20 with 1% FBS, were then added to wells and incubated for 2 hr at RT. Plates were washed 4 times each with PBST, PBS, and ddH$_2$O, developed with 100 µl/well BCIP/NBT (Calbiochem, Millipore) for 5–15 min in the dark, washed with ddH$_2$O and dried in the dark. ELISPOTS were quantified using Cellular Technologies LTD plate-reader and results analyzed using Cellspot software.

## BCR crosslinking and phosphorylation of signaling and adaptor proteins

Whole PBMCs were thawed and plated on a 96 well plate with $2 \times 10^6$ cells of each donor per well, and stained for CD10, CD19, CD20 CD21, CD27 and FCRL5 (509f6) (Biolegend) at 4°C in 4% PBS-FBS for 20 min. Cells were washed in 0.5% PBS-BSA and incubated at 37°C for 30 min before adding F(ab') 2 anti-IgM and anti-IgG (Southern Biotech, Birmingham, AL and Jackson ImmunoResearch, respectively) at a final concentration of 10 µg/ml and incubating at 37°C for 5 min. For the detection of phospho-proteins by flow cytometry, cells were fixed and permeabilized according to the manufacturer's protocol using the FoxP3 Staining Buffer Set (eBioscience). Cells were then stained with rabbit Abs specific for phospho-Syk (Y352) (pSyk) and phospho-PI3Kinase p85 (Y458)/p55 (Y199) (pPI3K) (Cell Signaling, Danvers, MA). Rabbit Abs were detected with AF 488-conjugated goat anti-rabbit IgG (Jackson ImmunoResearch). AF 647-conjugated mouse mAb specific for pPLCγ2 (Y759) and PE conjugated mouse mAb specific for BLNK (Y84) (pBLNK) were also used (BD Biosciences).

## Proliferation analysis

Isolated $2 \times 10^5$ naïve B cells, classical and atypical MBCs were incubated with an AF647- labeled proliferation dye (eBioscience) at 0.5 µM concentration for 8 min at RT. Cells were then washed twice in 15 ml of warm RPMI with 10% FCS and incubated for 4 days in the presence of IgG/IgM F(ab')2- and CD40-specific Abs (10 µg/ml) plus IL-10 (25 ng/ml), IL-4 25 (25 ng/ml) and CpG oligodeoxynucleotide type B (2.5 µg/ml).

## Measurement of cytokines in supernatants of stimulated B subpopulations

Supernatants of $10^5$ purified naïve B cells, classical and atypical MBCs were immediately analyzed after stimulation with LPS (Sigma-Aldrich) (100 ng/ml) or IgG/IgM F(ab')2- and CD40-specific (10 µg/ml) Abs plus CpG (2.5 µg/ml) for 12 hr. As per the manufacturer's instructions Bio-plex human cytokine assays (Bio-Rad Laboratories, Inc.) were used to detect the following cytokines: IL-6, IL-8 and CCL4. 50 µl of supernatant was incubated with anti-cytokine antibody-coupled magnetic beads for 30 min at RT shaking at 300 RPM in the dark. Between each step the complexes were washed 3 times in wash buffer, using a Bio-plex Pro II Station Wash (Bio-Rad Laboratories, Inc.). The beads were then incubated with a biotinylated detector antibody for 30 min before incubation with streptavidin-phycoerythrin for 30 min. Finally, the complexes were resuspended in 125 µl of detection buffer and 50 beads were counted with a Luminex 200 device (Bio-Rad Laboratories, Inc.). Final concentrations were calculated from the mean fluorescence intensity and expressed in pg/ml using standard curves with known concentrations of each cytokine.

## Calcium influx

Whole PBMCs ($2 \times 10^6$ cells) were stained for CD19, CD21 and CD27 and analysed for calcium influx using the Fluo-4 NW Calcium Assay kit (Invitrogen, Life Technologies). Briefly, a stock solution of the Fluo-4 dye was prepared by re-suspending the powdered dye in 10 ml of the provided assay buffer and 100 µl of 250 mM probenecid stock solution. Cells were re-suspended in 200 µl of the Fluo-4 solution for 30 min at 37°C in 5% CO$_2$, followed by incubation for 15 min at RT before analysis by FACS. Cells were analyzed on an LSRII for 25 s to establish a baseline, and the cells were then briefly removed from the FACS machine and stimulated with anti-IgM (10 µg/ml) and anti-IgG (10 µg/ml) (Southern Biotech and Jackson ImmunoResearch, respectively). The cells were mixed and immediately placed back on the FACS machine and analyzed for 150 s.

## Statistics

Analyses of next generation sequencing data of the IgH repertoire, gene expression microarray data and quantitative RT-PCR data are described above. Continuous data were compared using the paired or unpaired Student's T-test. ANOVA with Tukey's multiple comparisons test was also used to compare continuous variables. Spearman's correlation coefficient was used to examine the correlation

between continuous variables. The statistical test used is specified in figure legends. Statistical tests were computed using R version 3.1.2 (http://www.R-project.org) or GraphPad Prism version 5.0d (http://www.graphpad.com/scientific-software/prism/).

## Study approval

The Ethics Committee of the Faculty of Medicine, Pharmacy, and Dentistry at the University of Sciences, Techniques, and Technologies of Bamako, and the Institutional Review Board of the National Institute of Allergy and Infectious Diseases, National Institutes of Health approved this study. Written informed consent was received from participants prior to inclusion in the study. Written informed consent was obtained from parents or guardians of participating children prior to inclusion in the study.

## Acknowledgements

We thank the residents of Kambila, Mali for participating in this study. We thank Kishore Kanakabandi and Charles Turner for QPCR support and validation. We also thank Sarah Anzick for RNA extraction and target preparation. Biotinylated antibodies specific for FcRL4 and FcRL5 were kindly provided by Dr Randall Davis' laboratory at the University of Alabama at Birmingham. This work was supported by the Division of Intramural Research, National Institute of Allergy and Infectious Diseases, National Institutes of Health.

## Additional information

### Funding

| Funder | Grant reference | Author |
|---|---|---|
| Division of Intramural Research, National Institute of Allergy and Infectious Diseases | Z01 AI000949-02 LIG | Peter D Crompton |
| National Institutes of Health (NIH) | | Peter D Crompton |

The funders had no role in study design, data collection and interpretation, or the decision to submit the work for publication.

### Author contributions

SP, Conception and design, Acquisition of data, Analysis and interpretation of data, Drafting or revising the article; CMT, SFP, Acquisition of data, Analysis and interpretation of data, Drafting or revising the article; HS, YK, JW, SL, JS, KV, DES, SD, Acquisition of data, Analysis and interpretation of data; OKD, BT, Conception and design, Drafting or revising the article; KK, AO, Conception and design, Acquisition of data; IS, Analysis and interpretation of data, Drafting or revising the article; SKP, PDC, Conception and design, Analysis and interpretation of data, Drafting or revising the article

### Author ORCIDs

Jeff Skinner, http://orcid.org/0000-0001-5697-0442

### Ethics

Human subjects: The Ethics Committee of the Faculty of Medicine, Pharmacy, and Dentistry at the University of Sciences, Techniques, and Technologies of Bamako, and the Institutional Review Board of the National Institute of Allergy and Infectious Diseases, National Institutes of Health approved this study. Written informed consent and consent to publish was received from participants prior to inclusion in the study. Written informed consent and consent to publish was obtained from parents or guardians of participating children prior to inclusion in the study. NIAID IRB protocols 07-I-N141 or 06-I-N147.

## Additional files

### Supplementary files

• Supplementary file 1. Ex vivo DEGs from naïve B cells, classical and atypical MBCs from 20 Malian adults. Transacripts are significant if FDR-adjusted p-value < 0.05 and absolute fold change > 1.5. Transcript ID is the Affymetrix accession number.

• Supplementary file 2. A. Microarray expression and qRT-PCR values of 10 selected genes from naive B, classical and atypical MBC from 20 individuals. B. Spearman correlation values of Microarray and qRT-PCR expression data of 10 selected genes. C. TaqMan primers and probes for qRT-PCR expression data of 10 selected and 3 housekeeping genes.

• Supplementary file 3. Antibodies used in flow cytometry analysis.

### Major dataset

The following dataset was generated:

| Author(s) | Year | Dataset title | Dataset ID and/or URL | Database, license, and accessibility information |
|---|---|---|---|---|
| Portugal S, Tipton C, Sohn HW, Kone Y, Wang J, Li S, Skinner J, Virtaneva K, Sturdevant D, Porcella S, Doumbo OK, Doumbo S, Kayentao K, Ongoiba A, Traore B, Sanz I, Pierce S, Crompton PD | 2015 | Malaria-associated atypical memory B cells exhibit markedly reduced B cell receptor signaling and effector function | http://www.ncbi.nlm.nih.gov/geo/query/acc.cgi?acc=GSE65928 | Publicly available at NCBI Gene Expression Omnibus (GSE65928). |

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
