## [Decision Letter]

Thank you for sending your work entitled “Malaria-associated atypical memory B cells exhibit markedly reduced B cell receptor signaling and effector function” for consideration at *eLife*. Your article has been favorably evaluated by Tadatsugu Taniguchi (Senior editor) and four reviewers, one of whom, Urszula Krzych, is a member of our Board of Reviewing Editors.

The Reviewing editor and the other reviewers discussed their comments before we reached this decision, and the Reviewing editor has assembled the following comments to help you prepare a revised submission.

This study seeks to increase our understanding on the role aMBCs in response to persisting or chronic *P. falciparum* infection (with a major contribution from the current authors). The study is both timely and important in enhancing our understanding of the origins and function of aMBCs. Specifically, the differential expression of FcRL5 and FcRL3 is interesting and suggests that atypical memory B cells acquire different phenotypes dependent on the specific type of chronic infection. It will be interesting to understand the differences in the phenotype and function of atypical memory B cells induced as a result of different chronic infections in the future. In summary, the manuscript provides a detailed comparison of atypical and classical memory B cells isolated from individuals living in a high transmission area. The data are likely to lead to further investigation into the function of these cells both during and after chronic exposure to *P. falciparum* as well as other chronic infections. Although much has been reported previously by this and other groups, both in malaria and chronic viral infection, overall this is an interesting study and well conducted, analysed and presented.

1) The issue of antigen specificity is not addressed and it is quite imperative that results from experiments showing antigen-specific B cell responses are included.

2) The biochemical and molecular analyses are interesting but of limited scope. None of the experiments address the very interesting questions outlined in the Abstract regarding the potential relationship between aMBC and cMBC response to re-infection.

3) There was lower somatic hypermutations in B cells from malaria-exposed individuals compared to American adults.

A) How did these groups differ in age?

B) Would there be any difference between parasite negative and positive Malian adults? Although it is reported here that the Malian adults were asymptomatic, it is not clear if this is in terms parasite positivity, as asymptomatic simply means no symptoms.

4) Methodology:

A) Please indicate how many of either FACS or magnetic bead sorted cells were obtained and from what amount of blood, and how many cells were used for the specific experiments. This information must be included; otherwise it becomes difficult to appraise and critic the results presented. Furthermore, scientific methods ought to be presented in a way that others can replicate the experiments.

B) The purity of the sorted aMBC, cMBC, and naïve B cells ranged between 75–98%? Do the impurities include other B cell types? Was there a deliberate attempt to avoid plasmablasts?

C) Please include methods for proliferation.

D) Please justify and reference the use of TLR4 in human B cells stimulation.

E) Is LPS used for PBMC stimulation or is it targeted to the monocytes?

---

## [Author Response]

*1) The issue of antigen specificity is not addressed and it is quite imperative that results from experiments showing antigen-specific B cell responses are included*.

We agree that profiling the antigen specificity of atypical MBCs will ultimately be necessary to fully understand the biology of these cells. Importantly, we found that atypical MBCs do not actively secrete antibodies or differentiate into antibody-secreting cells when stimulated, thus precluding a direct assessment of their antigen specificity through conventional methods such as ELISA or ELISPOT.

In ongoing work we are taking two approaches to determine the specificity of atypical MBCs. First, we are optimizing flow cytometry protocols and reagents to directly quantify antigen-specific B cells by staining surface IgG with labeled antigen. Preliminary findings from our lab and collaborators indicate that currently available protocols and reagents lack the specificity required to generate interpretable data. In general, this approach has proved challenging for the field given the low frequency of any given antigen-specific MBC population in peripheral blood. In a second ongoing effort we are cloning and expressing antibodies from bulk-sorted atypical MBCs and classical MBCs to compare their specificity profiles using protein microarrays representing the entire *P. falciparum* proteome of over 5,400 gene products. We embarked on this unbiased but laborious approach because the malaria field has yet to identify unambiguous targets of protective antibodies.

As noted in the Introduction, several lines of evidence suggest that *P. falciparum* infection per se contributes to atypical MBC expansion including a positive correlation between atypical MBCs and *P. falciparum* transmission intensity; the differential expansion of atypical MBCs in age-matched children living under similar conditions in rural Kenya with the exception of *P. falciparum* exposure; and the appearance of atypical MBCs in the blood of healthy adults following experimental *P. falciparum* infection. Additionally, in the study described in our manuscript we found that malaria-associated CD21^–^CD27^–^-atypical MBCs express FcRL3 and FcRL5 but not FcRL4, in contrast to HIV-associated CD21^–^CD27^–^ exhausted MBCs that express FcRL4. This important finding suggests that the quality of CD21^–^CD27^–^ MBC expansion is not a generic response to chronic infection but varies by pathogen.

As noted by the reviewers, we anticipate that the conceptual advances in the current manuscript will likely stimulate others to further investigate the function and specificity of atypical MBCs both during and after chronic exposure to *P. falciparum* as well as other chronic infections. We added text to the Discussion to emphasize these important points.

*2) The biochemical and molecular analyses are interesting but of limited scope. None of the experiments address the very interesting questions outlined in the Abstract regarding the potential relationship between aMBC and cMBC response to re-infection*.

We examined the earliest events in B cell receptor signaling and the consequent downstream effector functions including proliferation, cytokine production and antibody production. The results of these experiments clearly point toward critical functional differences in atypical and classical MBCs. As the reviewers suggest, it will be of interest in future longitudinal studies to further compare the biochemical and molecular features of atypical MBCs and classical MBCs before, during and after malaria re-infection. We added text to the Discussion to underscore this important point.

*3) There was lower somatic hypermutations in B cells from malaria-exposed individuals compared to American adults*.

A) How did these groups differ in age?

In the current study we generated B cell somatic hypermutation data only from Malian adults. In the Discussion we contrasted these data with published B cell somatic hypermutation data generated by others in U.S. adults. Although this informal comparison hints at different somatic hypermutation rates, we opted to remove this reference from the Discussion because of important methodological differences between the studies. It will be of interest in future studies to conduct head-to-head comparisons of B cell somatic hypermutation rates in age-matched individuals with and without a history of exposure to malaria and other pathogens.

B) Would there be any difference between parasite negative and positive Malian adults? Although it is reported here that the Malian adults were asymptomatic, it is not clear if this is in terms parasite positivity, as asymptomatic simply means no symptoms.

All subjects included in the IgH repertoire analysis, which generated the somatic hypermutation data, were *P. falciparum* negative by thick blood smear. We added text to the Results section to clarify this point.

*4) Methodology*:

*A) Please indicate how many of either FACS or magnetic bead sorted cells were obtained and from what amount of blood, and how many cells were used for the specific experiments. This information must be included; otherwise it becomes difficult to appraise and critic the results presented. Furthermore, scientific methods ought to be presented in a way that others can replicate the experiments*.

This information has been added to the Methods and Results sections.

B) The purity of the sorted aMBC, cMBC, and naïve B cells ranged between 75–98%?

We added text to the Results section and representative flow cytometry plots to Figure 2—figure supplement 1 and Figure 3—figure supplement 1 to clarify the sorting methods and the levels of purity achieved. Specifically, the negative B cell selection kit removed all non-B cells and plasma cells/plasmablast and consistently yielded total B cells of >98% purity. These negatively selected B cells were separated into subpopulations either by FACS sorting or magnetic columns. For sorting, we used conjugated monoclonal Abs specific for CD10, CD19, CD20, CD21 and CD27 and achieved the following purity: naïve B cells 96.4%, classical MBCs 93.9% and atypical MBCs 91.5%. For magnetic isolation, we assessed purity by flow cytometry using conjugated monoclonal Abs specific for CD19, CD21 and CD27 and achieved the following purity: naïve B cells 93.2%, classical MBCs 73.0% and atypical MBCs 81.9%.

Do the impurities include other B cell types?

As described above, the first step in all experiments involving B cell subpopulations was the isolation of total B cells with a negative selection kit that consistently yielded purity of >98%. Therefore, contamination from non-B cells was negligible in all experiments. In the case of magnetic isolation, both classical MBC and atypical MBC contained a similar proportion of contaminating naïve B cells, which is expected given that naïve B cells comprise the largest subpopulation of total B cells. To clarify this point in the manuscript we added representative flow cytometry plots of purified subpopulations to Figure 2—figure supplement 1 and Figure 3—figure supplement 1 and edited the manuscript to clarify that the B cell subpopulations were obtained from highly purified B cells.

Was there a deliberate attempt to avoid plasmablasts?

Yes, we deliberately removed plasmablasts unless the experiment required plasmablasts as a positive control for Ab-secreting cells (Figure 6). As described above, all experiments that involved sorted B cell subpopulations began by removing all non-B cell and plasma cells/plasmablast from PBMCs using a kit that contained Abs specific for CD2, CD3, CD14, CD16, CD36, CD43, CD56, CD66b and glycophorin A, thereby removing plasma cells/plasmablasts that express CD43 and CD66b. Figure 6 shows that the removal of plasma cells/plasmablasts was highly effective.

*C) Please include methods for proliferation*.

We added a description of the proliferation assay to the Methods section.

*D) Please justify and reference the use of TLR4 in human B cells stimulation*.

To be convinced that atypical MBCs could not be activated to secrete Abs we tried a number of conventional B cell stimulants. Because these failed to activate atypical MBCs we considered alternatives. Although TLR4 is expressed at low levels on human resting B cells, it is upregulated on B cells in individuals with inflammatory conditions ([39]; [26], and [15]) and we considered that atypical MBCs may respond to LPS. We added text and references to the Results section to justify the use of LPS to stimulate human B cells.

E) Is LPS used for PBMC stimulation or is it targeted to the monocytes?

As described above in response to 4B all experiments, including those involving LPS stimulation, used B cells that had been negatively selected with a kit that consistently yielded purity of >98%. Therefore, contamination from non-B cells, including monocytes, was negligible.